# Elucidating the Structural Impacts of Protein InDels

**DOI:** 10.3390/biom12101435

**Published:** 2022-10-07

**Authors:** Muneeba Jilani, Alistair Turcan, Nurit Haspel, Filip Jagodzinski

**Affiliations:** 1Department of Computer Science, University of Massachusetts Boston, Boston, MA 02125, USA; 2Department of Computer Science, Western Washington University, Bellingham, WA 98225, USA

**Keywords:** computational structural biology, protein InDel mutations, graph-theory, rigidity

## Abstract

The effects of amino acid insertions and deletions (InDels) remain a rather under-explored area of structural biology. These variations oftentimes are the cause of numerous disease phenotypes. In spite of this, research to study InDels and their structural significance remains limited, primarily due to a lack of experimental information and computational methods. In this work, we fill this gap by modeling InDels computationally; we investigate the rigidity differences between the wildtype and a mutant variant with one or more InDels. Further, we compare how structural effects due to InDels differ from the effects of amino acid substitutions, which are another type of amino acid mutation. We finish by performing a correlation analysis between our rigidity-based metrics and wet lab data for their ability to infer the effects of InDels on protein fitness.

## 1. Introduction

A common type of mutation in protein structures are amino acid insertions and deletions, or InDels. InDels are caused by non-frameshifting insertion or deletion variants in the genome [1,2]. While another type of mutation, amino acid substitutions, has been the focus of a good deal of research, InDels are much less understood and studied. This is primarily due to the cost of wet-lab experiments [3,4], which require performing insertions or deletions at the sequence level followed by transcription and translation, resulting in a protein structure, the mutant, which differs from the wildtype and often cannot be modeled [5]. From a structural perspective, an InDel happens when a non-frameshift (NFS) insertion or deletion in the DNA sequence causes one or more amino acids to be inserted (Figure 1a) or deleted (Figure 1b) from the protein sequence when compared to the wildtype. The resulting InDel structure contains one or more inserted or deleted residues. The DNA InDels that are responsible for protein InDels may occur due to many reasons. Sometimes the cause is genome duplication. DNA InDels can also be due to replication errors or proliferation of transposed elements [6,7].

The location of an InDel can be anywhere in the sequence of a protein but frequently they occur in the loop regions [8]. Available data indicate that InDels can also occur in secondary structures, and these InDels are known to cause greater structural and functional changes compared to their loop counterparts [9]. However, InDels on loops can still have structural or functional effects—for example, the F508del of the cystic fibrosis transmembrane conductance regulator (CFTR) is one of the most common mutations that causes cystic fibrosis [10]. Mutations are oftentimes responsible for causing changes in the function of a protein [11]. It has been demonstrated that compared to substitutions, InDels are more correlated with functional changes in proteins [12,13]. In summary, there is substantial evidence that InDels are among the main factors when it comes to structural changes in proteins.

Research indicates that many disease phenotypes are caused by InDels. An example is the widely studied cystic fibrosis, where several mutants are known [14]. The F508del mutation in nucleotide-binding domain-1 (NBD1) of the cystic fibrosis transmembrane conductance regulator (CFTR) is the predominant cause of cystic fibrosis [15]. InDels are also linked to several types of cancer [7,16]. Moreover, severe acute respiratory syndrome coronavirus 2 (SARS-CoV-2) has variants that are caused by InDels [17]. Investigations of the spike protein of this virus reveal that InDels at the S1/S2 subunits result in mutants which are more resistant to vaccines [18].

The size of an InDel is found to correlate positively with the impact and extent of the changes to the structure of a protein. Predominantly, InDels that involve two or more more amino acids are found to have a greater impact on a protein’s structure whereas a single InDel tends to have a less pronounced effect [19]. Since the data on InDels is limited, there is a gap in research on the structural effects of InDels. A comprehensive study on the topic could have significant implications in protein modeling. In this work, we analyze the effects of both single and multiple InDels on the rigidity of a protein. We study these effect both local to the InDel region, and on a global scale, in order to attain a comprehensive understanding of how InDels effect the rigidity and flexibility of proteins.

InDels should also be examined relative to other mutations. InDels are thought to have a greater impact on the structure of a protein, but substitutions are the most common type of mutation, for which abundant experimental data is available. A comparison of what happens when a residue is inserted or deleted at a location versus when a residue is substituted for another, could offer valuable insights to the effects of InDels, as it is a near-direct comparison with a far more studied type of mutation.

### Previous and Current Work

Currently there are no well-established tools to model structural or functional changes in proteins caused by InDel mutations. Nor are there are any tools that systematically catalog mutants of a protein structure with InDels or substitutions. A previous effort, InDelPDB [20], used protein sequence alignments to detect InDels. SeqFIRE was another effort to extract conserved blocks and used entropy information to identify the locations of InDels and their conservation frequencies [21]. These tools are no longer available for public access and use.

In this work, we employ inverse kinematics, ab initio energy minimization and rigidity analysis, to extend our previous work to model and analyze InDels in protein structures [22]. The methodology employed in this work serves a two-fold goal. The first aim is to validate our ability to generate InDel mutants computationally and we do so by comparing the mutants with their PDB equivalents. The second goal is to explore in-depth the structure and stability properties of InDels mutants versus their wildtype counterparts in order to look into probable mechanisms to gauge the extent to which an InDel has an impact on the structure and rigidity of a protein. Further, we investigate the impact of substitutions when compared with InDels in order to compare the effects of various mutation types on the structure of a protein. In our previous work, we only assessed the effects of InDels in the loop regions, but here we also reason about insertions and deletions in secondary structures. In our previous work, we similarly assessed the effects of InDels only, but in this work we assess the effects of InDels when compared to substitutions in the same protein.

## 2. Methods

Our methodology consists of identifying wildtype-mutant pairs for InDels from the PDB [23] followed by generating mutants computationally. In the end, the computer generated mutants are compared with their PDB counterparts using a rigidity analysis methodology and statistical tests.

### 2.1. Identifying InDels in the PDB

Because there are no widely accepted methods to identify mutant structures in the PDB, this was our first task. We performed an expanded search on the PDB, looking for certain keywords, including *insertion*, *deletion*, and *mutation*, in the primary publications for protein structures, and we relied on UniProt [24] to identify PDB codes for wildtype and mutant proteins.

The PDB files containing InDels were further investigated in order to identify the location and the size of the mutation, where size refers to the count of amino acids that are inserted or deleted. This yielded a total of 35 mutants with InDels of size 1–6 in loop regions or in secondary structures. This was followed by using UniProtKB [25] to identify the PDB codes for wildtype proteins and InDel mutants. The product of this process was wildtype-mutant pairs. It should be noted that we limited ourselves to relatively short InDels since longer InDels may be difficult to model. In the future we plan to explore the modeling of larger InDels, possibly with the help of AlphaFold [26] or other new modeling techniques. It was recently found, however, that AlphaFold may also underperform on low confidence regions [27]. Therefore, we will also explore other refinement techniques.

### 2.2. Creating InDels and Refining the Resulting Structures

Computational mutants were generated from the wildtype protein PDB files using Rosetta [28]. It is a widely used software suite with a variety of molecular modeling applications. It provides multiple tools for predicting protein structures and also permits designing novel protein complexes. We used Rosetta for the purpose of local modeling and energy minimization.

Structural bioinformatics researchers have been utilizing robotics-based geometric algorithms for modeling and elucidating various properties of proteins. In proteomics, these algorithms are used to model protein structures and their dynamics [29].For motion planning, a protein structure is represented as a sequence of rigid bodies that are joined by elastic connectors [30]. In order to simulate protein motion, degrees of freedom (DoF) of the protein bond angles, bond lengths, and dihedral angles are considered and exploited. When end constraints are provided, the configuration of a chain is modeled using Inverse Kinematics (IK) [29].

One of the uses of IK in protein structure prediction is for modeling of loops by manipulating the rotational DoF of the loop region in order to find likely loop conformations that attach to the rest of the protein structure. An idea branched from IK is kinematic closure (KIC) (Figure 2). KIC is employed for computing probabilistic arrangements of a set of objects that are bound by common constraints.

The kinematic closure [31] process starts off by the division of the missing region into three pivots showed in green at top of Figure 2. The black dotted line in Figure 2 signifies storing the rigid body transformation from start till the end of the loop region. After that, the DoF is perturbed and in the end, pivot torsion values are computed and each rigid segment is oriented in a way that the starting rigid body transformation is reinstated [31,32].

For a kinematic articulated chain with a gap, that represents deletion of an amino acid or a position where one or more amino acids are inserted, we need to solve the problem in a systematic manner. The solution entails finding a balance between global adjustments of the torsion angles as well as the modification of angles and regions next to the residues that were inserted or deleted. Our approach to resolve this problem is as follows:Close the loop using geometric modeling without applying a refinement step;Employ kinematic closure to refine the loop region.

The specific protocols used to achieve the above are Rosetta’s *remodel* protocol [33], used to close the gap at the onset, followed by the application of the *loop* protocol [34,35].

The explanation for this particular combination is that structures that are generated by simply closing the gap do not have good energy scores. The is due to the reason that the *remodel* protocol’s *quick and dirty* variant closes the gap but the resulting structures have steric clashes. To resolve the structure, the *loop* protocol uses KIC to refine the region of closure in order to find a low energy conformation. This approach yields structures with good energy scores. In some cases the resulting structures were further enhanced by using all-atom refinement using Rosetta’s *relax* [36] protocol.

In order to validate our ability to generate mutants computationally, we compared the structures of the PDB mutants with their respective in silico counterparts. We performed the evaluation based on the RMSD of the loop region or the secondary structure where the InDel existed and computed all-atom RMSD before performing rigidity analysis of the mutant structures.

In Figure 3a, the left panel illustrates the process of deleting PRO-96 and TRP-97 for the mouse monoclonal antibody 1a7n where the deletion is encircled in black, and the right panel shows the computationally generated mutant superimposed on the experimentally available structure (PDB:1a7o). The global all atom RMSD is 0.283 Å, and the local RMSD score is 0.251 Å.

Previous research shows that InDels are more tolerated in the loop region of a protein [9]. However, there is evidence that InDels can be tolerated in the secondary structures as well [37]. The existing literature shows that insertions are tolerated more within the secondary structural elements compared to deletions [19]. Our current scripts employ Rosetta to insert and delete the amino acids virtually anywhere in the protein.

The methodology to perform insertions and deletions in secondary structures differs from loop regions as follows:Close the gap without refinement of resulting structure using geometric modeling.Perform all-atom refinement on the resulting structure for the purpose of energy minimization.

In Figure 3b, the left panel illustrates the process of insertion of ARG-34 for the *Escherichia coli* thioredoxin (PDB: 1txx) where an arrow is pointing towards the region where insertion is performed. The right panel shows the computationally generated mutant superimposed on the experimentally available structure (PDB:1tho). The global all atom RMSD is 0.685 Å, and the local RMSD score is 0.221 Å. Again Rosetta’s *remodel* protocol was used to close the gap and then Rosetta’s *relax* protocol was used in the second step to resolve the structures for achieving better energy scores. The rationale for this particular combination is that using KIC on the secondary structure did not yield structures similar to the PDB mutants. However, all-atom refinement results in structures that are more similar to their PDB counterparts.

### 2.3. Rigidity Analysis

The rigidity and flexibility of protein domains provide insights into a protein’s structural stability [38]. A computational analysis of the rigidity of a protein can yield insights into the effects of amino acid substitutions [39,40]. We relied on the software suite KINARI [41] to assess the rigidity of our InDel mutants. KINARI takes a PDB file as input, identifies stabilizing interactions such as hydrogen bonds, models the protein as a Body–Bar–Hinge Framework, and runs a Pebble Game analysis on an associated graph representing the mechanical model of the Body–Bar–Hinge Framework. The output for rigidity analysis is a list of atoms that exist among identified rigid clusters. While rigidity analysis does not directly measure the functional effects of mutations on a protein—only the structural effects—we have shown in the past, through our single and later multiple mutation studies, that we can correlate rigidity properties with function [39,40,42]. In particular, our approach predicted the change to the free energy of unfolding upon mutation ΔΔ*G* for 2700 single and multiple point mutations, using a combination of the rigidity analysis metrics (RDSMs) and structural features that include solvent accessible surface area (SASA), temperature, pH, secondary structure element, and the type of mutated amino acid. We used these as features for Support Vector Regression (SVR), Random Forest (RF), and Deep Neural Network (DNN) methods. Our results showed high correlation (0.81) between rigidity and ΔΔ*G*. Finding and correlating ΔΔ*G* information with rigidity for InDels is more challenging, due to the dearth of experimental data. However, there exists some information about specific families, for example luciferase [43], TEM-1 β lactamase [1] and green fluorescence protein [44] among others. Many of the existing works either measures the effect of InDels on sequences or use other measures of fitness [1]. Since little ΔΔ*G* information exists, we used other fitness measures for baseline validation of our results. Section 3.8 provides the result of our baseline validation.

We measured the differences between the in silico generated InDel mutants and their wild types, and between the PDB InDel mutants and the wildtype, using RMSD measurements, and via a visual inspection via PyMol. The similarities and differences in these scores give us insights into the ability of our method to accurately generate in silico InDels.

### 2.4. Rigidity Analysis to Measure InDel Effect

For a quantitative analysis of the structural differences between our wildtype and InDel mutants, we used our previously developed Rigidity Distance Similarity Metric (RDSM). It tells us information about the sizes and amounts of identified rigid clusters in the wildtype when compared to the generated mutant [42]. For RDSM:$$\mathrm{RDSM}=\sum_{i=1}^{x=LRC}i\times w\left(x\right)\times \left[WT_{i}-Mut_{i}\right]$$
*i* is the rigid cluster size, and *LRC* = Largest Rigid Cluster in the wild type. $WT_{i}$ and $Mut_{i}$ are the count of rigid clusters of size *i* in the wildtype and mutant, respectively. This provides a measure of the accumulated differences between the sizes of each protein’s rigid clusters. If the wildtype of a protein does not have a rigid cluster of size *i*, but the mutant does, then that particular term in the summation is negative. Conversely, if the wildtype but not the mutant, has a rigid cluster of size *i*, then that term in the summation is positive. The term w(x) is a weight factor, defined by the following:$$w\left(x\right)=\frac{1}{1+e^{-0.1x+5}}$$

The RDSM measurement was developed in order to quantify differences in rigidity properties between two proteins. It accounts for the difference in the size and numbers of rigid clusters, while weighing the difference between the largest clusters in the mutant and wild type more heavily than the differences among the smallest clusters. This is because differences among the largest rigid clusters are usually considered to have a bigger effect on protein rigidity than small clusters, which have a more localized effect.

We calculated the RDSM scores for each of the wildtype–experimental mutant, wildtype–computational mutant, and experimental mutant–computational mutant pairs. This allows us to evaluate the quality of our computational approach for creating these InDels. Lower RDSM scores mean the rigid clusters of a given wildtype and mutant are more structurally similar to one another. Due to variations in protein size, the RDSM scores were normalized by the number of atoms in that protein. This lets us better assess the differences as relative and not absolute.

In addition to the RDSM metric, we computed the Two Largest Cluster Comparison Score (TLCCS) [45] between the wildtype and PDB mutant. TLCCS assesses the difference between the size and overlapping atoms of the largest two clusters of two proteins—in our case, the wildtype and the PDB mutant. TLCCS is calculated by:$$\mathrm{TLCCS}=\frac{\frac{\mathrm{AALC}\;\cap \;\mathrm{CGLC}}{\mathrm{AALCS}}}{1+\frac{\left|\mathrm{AALCS}-\mathrm{CGLCS}|\;+\;|\mathrm{AASLCS}-\mathrm{CGSLCS}\right|}{\mathrm{AALCS}}}$$
where AA = All-Atom rigidity analysis, CG = coarse-grained rigidity analysis, LCS = Largest Cluster Size, and SLCS = Second Largest Cluster Size.

Its particular use is in providing a quantitative score of how much structural change is caused by the mutation, because modeling more impactful mutations is of greater priority than modeling mutations that had little effect. The closer the TLCCS value is to 1, the less effect the mutation has on the protein’s rigid clusters. InDels that result in a TLCCS of near 0.5 and below will be treated as a having a significant impact in the protein’s rigidity.

Lastly, we performed visual analysis via a custom Python program, focusing on the rigid clusters at the InDel site. This elucidates important information about the effect on the protein’s rigid clusters in the proximity of the InDel site(s), providing new information that global analysis does not offer.

## 3. Results and Discussion

### 3.1. Baseline Similarity Score Validation

In order to further validate our rigidity analysis metrics in capturing the structural properties of proteins, we tested how small changes to a protein structure affect its rigidity properties. To do this, we subjected six protein structures from our data set to a small number of minimization steps using NAMD [46]. We created five variants of each protein by running 100, 200, 300, 400, and 500 energy minimization steps. The goal was to slightly perturb the structure but not induce large structural changes. We calculated the RDSM values of the variants compared to the original protein, as well as the scores depicted in Figure 4a,b (see more statistical analysis in Section 3.2 and Section 3.5). Our goal was to showcase the small differences in these scores between the wildtype and the variants, as well as between the computed mutants and the PDB mutants, which are also two variants of the same protein. Appendix A, show the outcome of this validation, including the numerical data for our NAMD generated variants. As can seen, the RDSM scores are in general rather small, as expected due to the small changes in the structure, and in most cases are much smaller than the RDSM scores for the wildtype-mutant pairs. This, in addition to our previous work [40,42], show that the RDSM scores can capture structural differences in proteins, and are robust to small structural changes.

### 3.2. Comparing PDB and Computed InDel Mutants

Table 1 shows the all-atom RMSD and RMSD local to the InDel region modeled between the PDB InDel mutant and our in silico generated InDel mutant. Here local RMSD signifies loop region or secondary structure RMSD of computational mutant versus PDB mutant.

Some of the entries refer to varying-length InDels from the same protein. For instance, 6AIS and 6ICS represent deletions of two and four amino acids, respectively, from the loop region of outer surface protein A of *Borreliella burgdorferi*. In all, the table represents 25 examples of deletions and 11 insertions. The maximum number of inserted residues is three. We were able to insert up to four residues in the loop region of the human lysozyme, but we were unable to find any preexisting InDel mutants in the PDB that contained more than three amino acid insertions. Three of the deletions are of length 6. The letters “l”, “s”, or “h” in the table represent whether the InDel exists in the loop region, beta sheet, or alpha helix of protein, respectively.

Both the global and local RMSD values, in angstroms, are all less than 1.0 Å, with several as low as 0.13 Å. Therefore our in silico approach for creating InDel protein mutants using the robotics-inspired inverse kinematics approach and ab initio energy minimization yields structures that are similar to InDel mutants whose structures are resolved experimentally.

### 3.3. Using Rigidity Analysis to Measure the Effect of an InDel

Using our RDSM scores, we determine the similarity between the effects of our computational InDel versus the experimental one. Previous research [42] found that two RDSM metrics, differing in their w(x) functions (Section 2.4), produced results best correlated with the established effects of substitutions:$$\mathrm{RDSM}2:w\left(x\right)=\frac{1}{1+e^{-0.1x+5}}$$
$$\mathrm{RDSM}3:w\left(x\right)=\frac{1}{1+e^{-0.05x+5}}$$

These scores, RDSM2 and RDSM3, for the wildtype, PDB, and in silico generated mutants, are shown in Figure 4a,b, respectively. The variation in rigidity properties between each of the pairs in each triplet is demonstrated in these figures. When the PDB InDel and in silico generated InDel have similar amounts of change on the protein’s resulting structure, we can assume the RDSM between both the wildtype–PDB mutant (blue bar) and wildtype–in silico mutant (orange bar) to be of similar size. If our modeling is in fact correct, we can also expect the RDSM between the PDB mutant and in silico InDel (gray bar) to be small, meaning little difference between them. In summary, we would expect both the PDB mutant and computational mutant to be of similar difference from the wildtype, and for the PDB mutant and computational mutant to be of little difference to each other.

### 3.4. Evaluating the Magnitude of InDel Effect

One initial shortcoming of our RDSM approach, is that it fails to show the magnitude of the structural change of the InDel in the real world. This information is important, because computationally modeling InDels that have little to no effect on a protein’s structure would not be as useful as modeling an InDel that fundamentally changes it. Consequently, we calculated the TLCCS between each wildtype and PDB mutant, with the results in Table 1. One thing that can be noticed, is that the values tend to be greater than 0.5, meaning that most InDels have a small structural effect when assessed via this metric. However, there are several instances in which the values are around or below 0.5, meaning a significant change in the protein’s rigid clusters has occurred, such as with 2BBO (cystic fibrosis), 2NIP (nitrogenase), and 1ANF (maltodextrin binding protein). The results of our computational pipeline is of greater interest when modeling such InDels as these because they have substantial impact on the structure of the protein.

### 3.5. Statistical Validation

We carried out a statistical analysis of the results in order to evaluate the choice of RDSM metric as an equitable approach to gauge the impact of an InDel. We describe the problem as follows: Let *X* indicate the RDSM score between wildtype and PDB mutant and let the RDSM score between the wildtype and computational mutant be represented by *Y*. This is indicated by orange and blue bars in the graphs, respectively. We used Spearman correlation test [47] in order to see the similarity among the two.

The evaluation was conducted for five RDSM values. These values varied in their weight w(x) value. For RDSM1 through 5, we obtained values 0.87, 0.916, 0.76, 0.6, and 0.46 with statistical significance of p<0.05. The Spearman correlation test disclosed that both the variables signifying PDB mutant vs. wildtype and computational mutant vs. wildtype demonstrated adequate correlation with each other.

The other part of the analysis comprised of finding the relationship between *X* and *Y* with the RDSM score between the PDB mutant and the computationally derived mutant. The problem is now comprised of performing comparison of *W* with *X* and *Y*. Which means that we could not simply use 2-sample *t*- or *z*-test. Thus, we used the subsequent approach to resolve this.

The first step was to compute the probability of the third variable being smaller than the minimum of other, i.e., P(W<min(X,Y)). This was performed by computing the difference between the minimum of *X* and *Y* versus *W*. Once the difference, denoted by variable *D*, is computed, the probability of it being less than zero was computed—i.e., D<0. We proceeded to repeat the same test for the RDSM values and attained the scores of 0.7638, 0.638, 0.6823, 0.623, and 0.6098 for RDSM 1–5, respectively. This served as an indicator [48] that the computational and InDel mutant are similar to one another as compared to the wildtype as per current data.

### 3.6. Effects of InDels Local to Mutation Site

Figure 5 demonstrates that even though the RDSM scores may not always show similarity, in this instance for the Enhanced Green Fluorescent Protein—wildtype 2Y0G and PDB mutant 4KAG—the RDSM score indicated there was a 2 fold greater effect on the wildtype from the PDB mutant than from the in silico mutant. When inspecting the local region, we can see our computational modeling did in fact create a quite similar effect to the PDB mutant. Another example, demonstrating the local-level similarity, is shown in Figure 6, using rHb1.1—wildtype 1C7C and PDB mutant 1C7D—while the effects of the InDel local to the mutation site are similar in both the PDB and in silico mutants, the rest of the protein is very similar as well. Most large rigid clusters were shrunken in both mutants, in favor of the very largest rigid cluster becoming larger. As seen in Table 1 and Figure 4a,b, there does not seem to be a big difference between structures where InDels occurred on loops and secondary structures, neither with respect to the RMSD nor with respect to the differences between the rigidity properties of the wildtype and the InDel. However, this may be due to the small size of the InDels. In the future we plan to investigate whether, and to what extent, InDels in loops and secondary structure elements have an impact on the protein’s structure.

### 3.7. Comparing InDels and Substitutions

To further investigate the effects of InDel mutations, they should be compared with existing mutations. In Figure 7, we show the TLCCS between each possible pair of the wildtype, substitution, and InDel for various proteins. As can be seen in the graph, in most instances the wildtype–InDel (orange bar) has a TLCCS equal to or lesser than that of the wildtype–substitution (blue bar). This would indicate that InDels typically have a greater effect than substitutions, in line with current knowledge [12]. However, in three instances, 1MDQ, 103L, and 215L, the substitution had a greater impact on the structure of the protein than the InDel did. These results suggest that InDels tend to have a greater effect than substitutions on a protein’s structure, but both types are still impactful.

### 3.8. Correlating TLCCS and RDSM Scores with Experimental Data

To assess the predictive power of our rigidity-based approach to infer the effects of InDels on protein structure, we applied our method on TEM-1 β-lactamase, an antibiotic resistance determinant, for which fitness effects of single amino acid InDels are reported in a study that relied on AMP resistance [1,49]. We generated InDel mutants of PDB file 1zg6, by randomly selecting 10 residue locations, and performed an in silico insertion using our approach. We recorded our TLCCS and various RDSM metrics, and assessed the correlations of these metrics against the experimentally derived fitness scores of the 10 InDel mutants (Table 2).

As can be seen, the Pearson Correlation Coefficient R values for RDSM1, RDSM2, RDSM3, and RDSM5 of −0.423, −0.191, −0.153, and −0.2547 reveal a negative weak correlation between these rigidity-based metrics and the experimentally derived fitness values for the 10 InDels. Although the correlations for the rigidity-based metrics are not strong, they nonetheless reveal that there is a relationship between the experimental fitness values and rigidity metrics for these 10 mutants.

In our previous study involving single- and multiple-substitution only mutations, RDSM1, RDMS2, RDSM3, and RDSM5 metrics had much weaker negative correlation values of −0.053, −0.059, −0.063, and −0.022 against ΔΔ*G* [40]. For that work, when the RDSM values were incorporated with various additional structural metrics such as charge and aromaticity of the mutant (substitution) residue, and secondary structure information where the substitution was performed, the ensuing machine learning model yielded a Pearson Correlation Coefficient of 0.8 for ΔΔ*G* against the rigidity-based prediction of the effect of the mutation. Thus, when combining rigidity metrics with other features of the InDel mutants, we can expect similar or better performance, which we leave for future work.

### 3.9. Case Study of InDel vs. Substitution Effect

Fully elucidating the effects of substitutions versus InDels is out of the scope of this project, but we provide a case study nonetheless. The Cypovirus Polyhedra protein has multiple crystal structures: PDB 5YR9 has a substitution at position 190, 5GQI has one deletion near that position, 5GQJ has two deletions, and 5GQN has three deletions. Figure 8 shows the difference between these 4 proteins at the site of their respective mutations. As can be seen, there is not a big difference between one deletion and one substitution. However, once we have multiple InDels as in 5GQJ and 5GQN, there is a clear visible change. This is in line with the conclusions found by Zhang et al. [19] where one InDel may not always have much effect, but multiple do. This one case study would imply a single InDel’s effects are not always very different from substitutions, but several InDels may provide unique effects that should be explored further.

## 4. Conclusions

When compared to substitutions, InDels are responsible for more structural and functional changes in proteins. In spite of that, they are not as well studied. One of the main reasons is the lack of experimental and computational data. In this study, we attempted to develop a better understanding of InDels as well as their structural implications.

In this work, we computationally generated short InDels, and also assessed their effects on a protein’s rigidity. For this study, we identified 36 InDels of lengths 1–6 amino acids, that were in the loop regions and secondary structures of protein. Those InDels were computationally generated, and then rigidity analysis was performed on the resulting structures. We demonstrated that the methods generate InDel mutants that are structurally similar to PDB InDel mutants.

The rigidity analysis of the computationally generated InDels proved that we are capable of producing InDels computationally with similar rigidity properties to the InDel mutant structures in the PDB, with respect to the size and location of rigid clusters and flexible regions.

Further, our statistical analysis revealed that computational and experimental PDB mutants both are significantly different from the wildtype while their rigidity properties are similar to one another.

This preliminary study highlights our ability to create computational InDel mutants, thus attempting to bridge the gap that exists due to lack of experimental and computational InDel data.

Furthermore, there is currently no widely known resource for identifying mutant proteins (InDels or substitutions) which mines the PDB for mutants of a desired protein. In our future work we plan to create a web-based tool that will directly mine the PDB for InDel and substitution mutant structures given a wildtype. Eventually we plan to incorporate a substitution mutant generator into a compute pipeline, which will permit us to create and analyze different types of protein mutants. Another future direction is to incorporate larger InDels and model them, possibly with the help of AlphaFold or other new modeling techniques.

## Figures and Tables

**Figure 1 biomolecules-12-01435-f001:**
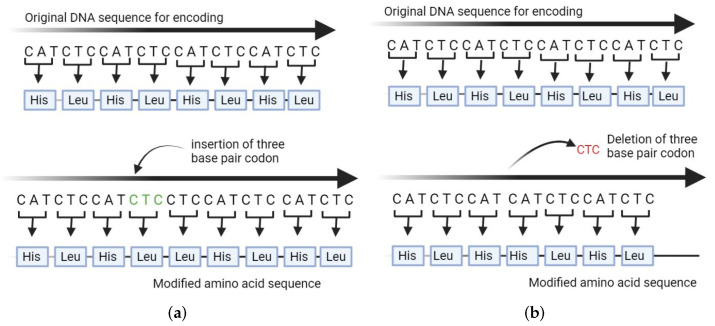
(**a**) An insertion mutation is caused by insertion of three base pair codons in the genome. (**b**) A deletion mutation is caused by deletion of three base pair codons in the genome.

**Figure 2 biomolecules-12-01435-f002:**
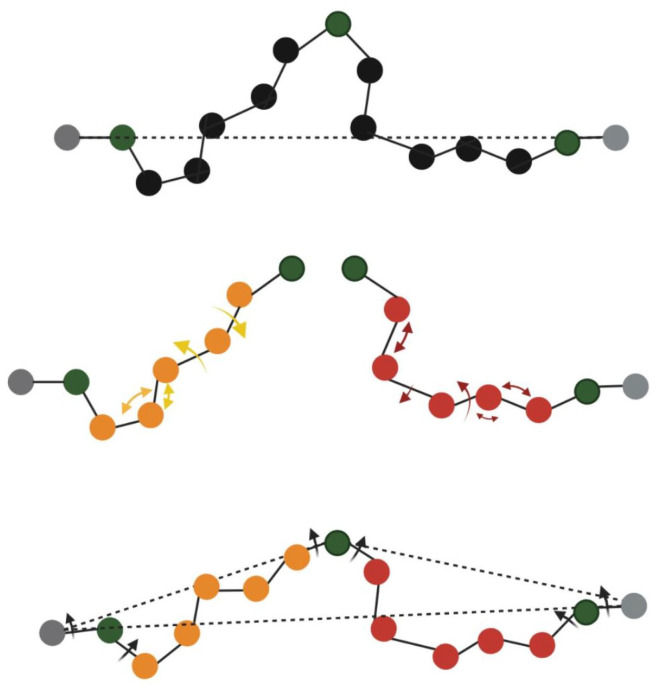
Kinematic closure illustrated in a top-down manner. The top panel demonstrates the computation and storing of rigid body transformations (black dotted line). The DoFs of bond lengths, torsions and bond angles are perturbed (portrayed by red and orange arrows in the middle panel). The third panel demonstrates that pivot torsion values are computed and stored and each rigid segment is oriented in such a fashion that original rigid body transformation is reinstated (black arrows).

**Figure 3 biomolecules-12-01435-f003:**
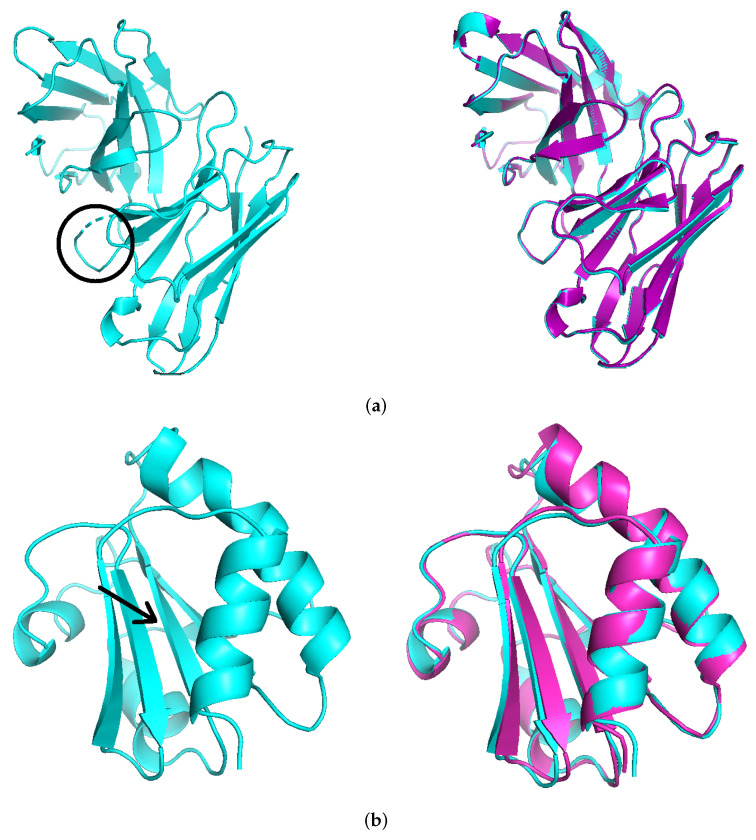
((**a**) **left**) 1a7n with PRO96 and TRP97 removed (encircled in black). (**right**) Computational mutant after fixing the gap and energy minimization superimposed on PDB 1a7o. ((**b**) **left**) 1txx with ARG34 inserted in a beta sheet (pointed by arrow). (**right**) Computational mutant (magenta) superimposed on PDB 1tho.

**Figure 4 biomolecules-12-01435-f004:**
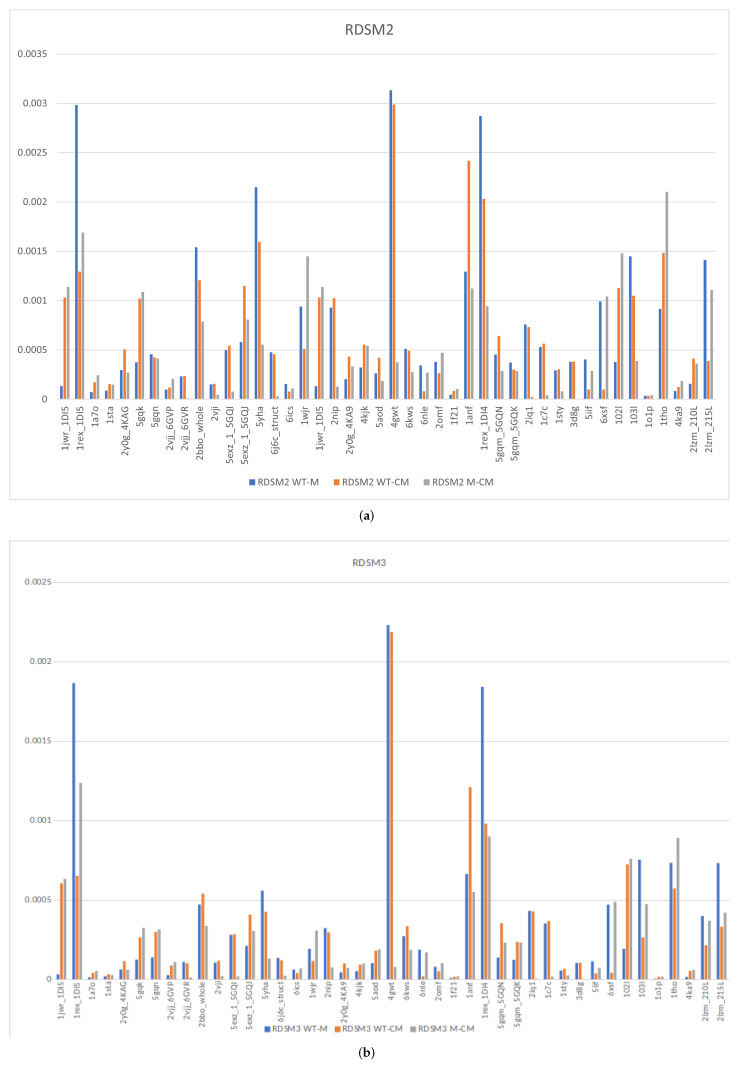
The results of calculating the RDSM2 (**a**) and RDSM3 (**b**) between each set of proteins. The blue bar shows the RDSM score between the wildtype and PDB Mutant, orange is the RDSM score between the wildtype and our computationally-generated mutant, and the gray bar is the RDSM score between the PDB mutant and computationally-generated mutant.

**Figure 5 biomolecules-12-01435-f005:**
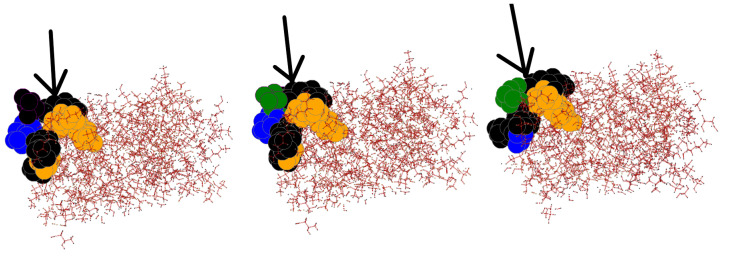
The 3D visualization of the rigid clusters of the Enhanced Green Fluorescent Protein (**left** to **right**): wildtype 2Y0G, PDB mutant 4KAG, and the computer-generated mutant. A deletion of aspartic acid at residue index 190 is modeled. The arrow points to the precise location of the InDel. The 5 largest clusters are colored, while all other clusters are black.

**Figure 6 biomolecules-12-01435-f006:**
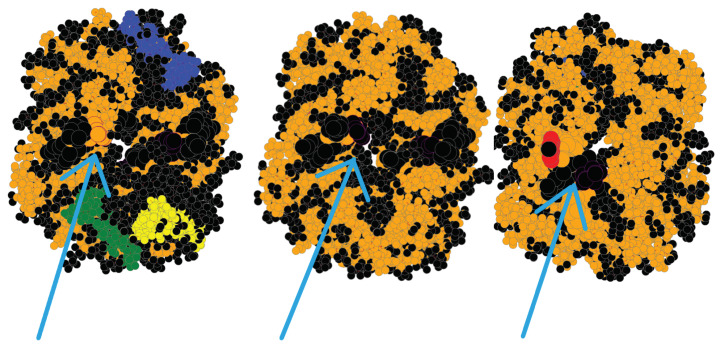
The 3D visualization of the rigid clusters of Deoxy rHb 1.1. (**left** to **right**) wildtype 1C7C, PDB mutant 1C7D, and the computer-generated mutant. A deletion of Glycine at residue index 143 is modeled. The arrow is drawn to the InDel’s precise location. The 5 largest clusters are colored, while all other clusters are black.

**Figure 7 biomolecules-12-01435-f007:**
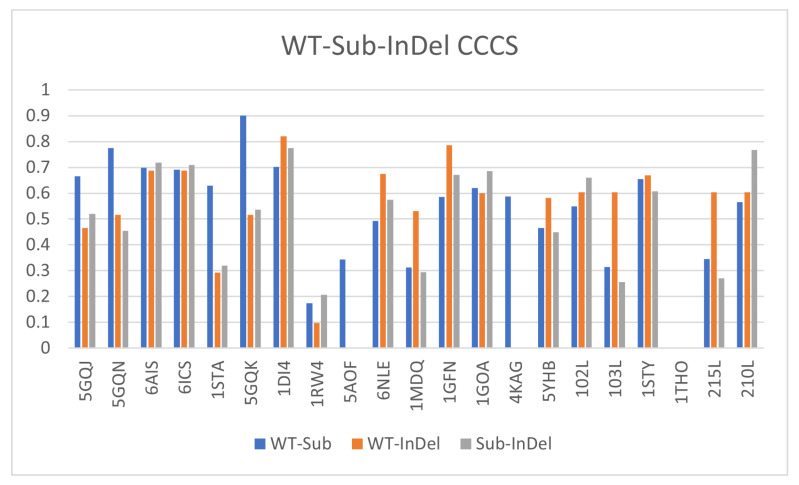
Graph showing the TLCCS between each pair of the wildtype–substitution (blue bar), wildtype–InDel (orange bar), and substitution–InDel (gray bar). Lower scores indicate a greater difference between their largest two rigid clusters.

**Figure 8 biomolecules-12-01435-f008:**
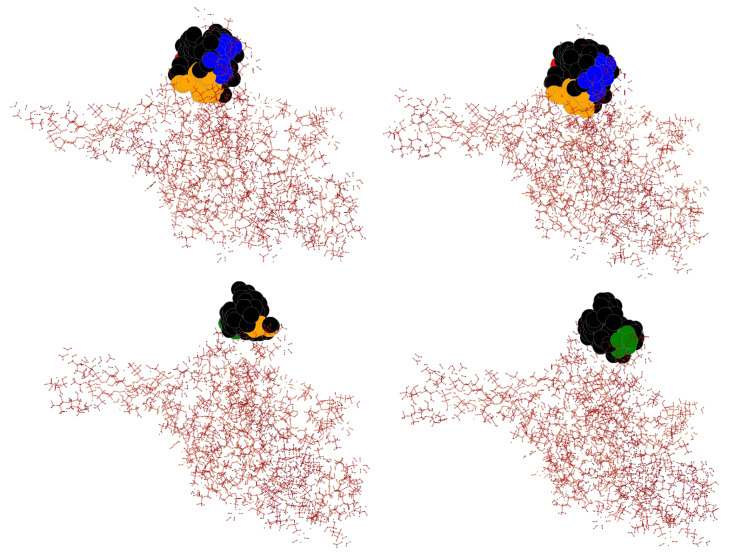
Visualization of the largest 5 rigid clusters at the site of the InDel/substitution for the Cypovirus Polyhedra proteins: 5YR9 with single substitution (**top-left**), 5GQI with single deletion (**top-right**), 5GQN with three deletions (**bottom-left**), and 5GQJ with two deletions (**bottom-right**).

**Table 1 biomolecules-12-01435-t001:** Global and local RMSD of PDB and in silico generated InDel mutants in angstroms. TLCCS provides a quantitative measure of the difference between the two largest clusters of the wildtype and PDB mutant. The lower the value, the bigger the difference, with values 0.5 or lower being significantly different.

Wildtype	Mutant	InDel Type	Global RMSD	Local RMSD	TLCCS
**5GQL**	5GQI	d:1, l	0.138	0.126	0.848
5GQL	5GQJ	d:2, l	0.767	0.708	0.665
5GQL	5GQN	d:3, l	0.836	0.678	0.774
6J6C	6AIS	d:2,l	0.96	0.509	0.698
6J6C	6ICS	d:4, l	0.135	0.34	0.691
2VJI	4XQF	d:2, l	0.16	0.135	0.765
**2BBO**	1XMJ	d:1, l	0.652	0.768	0.023
2IQ1	6AK7	d:3, l	0.79	0.392	0.693
2VJJ	6GVP	d:2, l	0.186	0.376	0.789
**1A7N**	1A7O	d:2, l	0.283	0.251	0.706
1STN	1STA	i:2, l	0.506	0.47	0.629
2VJJ	6GVR	d:2, l	0.168	0.329	0.790
5GQM	5GQK	d:3, l	0.796	0.269	0.899
1JWR	1DI4	d:2, l	0.563	0.498	0.54
1JWR	1DI5	d:1, l	0.283	0.251	0.81
2Y0G	4KA9	i:1, d:1, l	0.72	0.58	0.656
4KJK	4KJL	i:1, l	0.378	0.574	0.717
2NIP	1RW4	d:1, l	0.557	0.53	0.189
4EUL	6FLL	d:2, l	0.206	0.36	0.513
**1ANF**	1MDQ	i:1, l	0.701	0.607	0.312
1OMF	1GFN	d:6, l	0.245	0.342	0.554
1F21	1GOA	i:1, l	0.97	0.85	0.498
**2Y0G**	4KAG	i:1, d:1, l	0.69	0.74	0.587
5YHA	5YHB	d:3, l	0.623	0.807	0.460
**2LZM**	103L	i:3,hs	0.221	0.125	0.313
2LZM	102L	i:1, h	0.638	0.23	0.534
2EY6	6XSF	d:6, h	0.735	0.43	0.467
1EY0	5IIF	d:6, s	0.7	0.36	0.447
1SNC	1STY	i:1, h	0.709	0.385	0.671
**1C7C**	1C7D	i:1, h	0.745	0.35	0.775
**1TXX**	1THO	i:1, s	0.685	0.221	0.654
4EUL	4KEX	d:1, l	0.263	0.437	0.616
2LZM	210L	d:1, h	0.961	0.31	0.625
1C7D	1O1p	d:1, s	0.708	0.224	0.762
2LZM	215L	i:1, h	0.657	0.3	0.686

**Table 2 biomolecules-12-01435-t002:** Pearson Correlation Coefficient, R, values for TLCCS and RDSM metrics against experimentally derived fitness values for InDels TEM-1 β-Lactamase, PDB 1zg6. Insertion = residue number, amino acid inserted; Fitness = AMP resistance-based fitness value, taken from Appendix A, columns *Position* and *Fitness*, from [1]; TLCCS = Two Largest Cluster Comparison Score (Section 2.4); RDSM1, RDSM2, RDMS3, RDSM5: Rigidity Distance Similarity Metric Values for wildtype-mutant pairs (Section 2.4); R = Pearson Correlation, with *N* = 10 wildtype-Indel mutant pairs.

Insertion	Fitness	TLCCS	RDSM1	RDSM2	RDSM3	RDSM5
37, Q	0.164355	0.00051	0.655974	0.000121	0.000036	2.01 $\times \;10^{-6}$
45, P	0.000911	0.00062	0.662656	0.000146	0.000047	2.72 $\times \;10^{-6}$
89, V	0.022674	0.00046	0.651466	0.000086	0.000022	1.10 $\times \;10^{-6}$
130, C	0.001328	0.00033	0.672744	0.000109	0.000031	1.67 $\times \;10^{-6}$
148, K	0.237953	0.00008	0.669186	0.000097	0.000027	1.50 $\times \;10^{-6}$
150, V	0.305583	0.00011	0.697794	0.000062	0.000016	8.42 $\times \;10^{-7}$
172, M	0.435899	0.00043	0.676728	0.000128	0.000039	2.22 $\times \;10^{-6}$
191, G	0.005969	0.00041	0.676746	0.000117	0.000034	1.89 $\times \;10^{-6}$
210, G	0.028093	0.00036	0.698456	0.000099	0.000028	1.55 $\times \;10^{-6}$
214, I	0.083574	0.00056	0.673495	0.000123	0.000037	2.08 $\times \;10^{-6}$
	R	0.239	−0.423	−0.191	−0.153	−0.254

## Data Availability

The protein structures used the current study are available in the protein data bank (PDB) available at: https://www.rcsb.org/.

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
