# Peer review of "Elucidating the Structural Impacts of Protein InDels"

_biomolecules, 2022, doi:10.3390/biom12101435_

Round 1

Reviewer 1 Report (Previous Reviewer 3)

The authors have addressed the issues I raised in my last review and I'm agreeable to publication of the manuscript.

Author Response

This reviewer did not request edits or changes. Thank you for their suggestions in earlier review rounds, which we addressed.

Reviewer 2 Report (Previous Reviewer 2)

I am repeating the critique of my previous rejection. The pedictive power of the method must be tested against some experimental data. This need not necessarily be deltadeltaG, although I would consider it the most obvious. Function (enzymatic activity of an enzyme, fluorescence of GFP), solubility, foldability might be other useful parameters available from literature.

l. 196: The authores state "While we do not currently have deltadeltaG information for short InDels, our past results are encouraging." This information must be out there, although it might be cumbersome to gather. Limited literature search already reveals a number of sources:

Nat Commun. 2021; 12: 3616.
Engineering the protein dynamics of an ancestral luciferase
deltadeltaG of 25 InDel mutants in a bar chart

J Proteome Res; 2019 Mar 1;18(3):1402-1410.
Analyzing Change in Protein Stability Associated with Single Point Deletions in a Newly Defined Protein Structure Database
Anupam Banerjee, Yaakov Levy  1 , Pralay Mitra
Software for Single Point Deletions prediction for stability changes

PLOS ONE | https://doi.org/10.1371/journal.pone.0164905 April 3, 2017
Computational prediction of the tolerance to amino-acid deletion in green-fluorescent protein
Eleisha L. Jackson1,2,3, Stephanie J. Spielman4, Claus O. Wilke1,2,
weighted contact number (WCN) is a main predictor for deletion tolerance

Structure, Volume 22, Issue 6, 10 June 2014, Pages 889-898
Random Single Amino Acid Deletion Sampling Unveils Structural Tolerance and the Benefits of Helical Registry Shift on GFP Folding and Structure
James A.J.Arpino13Samuel C.Reddington1Lisa M.Halliwell1Pierre J.Rizkallah2D. DafyddJones1
87 tolerated and non-tolerated single aa deletions in EGFP, DeltaDeltaG for EGFP_G4Delta, EGFP and EGFP_G4Delta structures in PDB

Fitness Effects of Single Amino Acid Insertions and Deletions in TEM-1 beta-Lactamase.
Gonzalez CE, Roberts P, Ostermeier M. J Mol Biol. 2019 May 31;431(12):2320-2330. doi: 10.1016/j.jmb.2019.04.030. Epub 2019 Apr 26. PMID: 31034887 Free PMC article.
large list of fitness effects of InDels in beta-lactamase

Structural plasticity of green fluorescent protein to amino acid deletions and fluorescence rescue by folding-enhancing mutations.
Liu SS, Wei X, Dong X, Xu L, Liu J, Jiang B. BMC Biochem. 2015 Jul 25;16:17. doi: 10.1186/s12858-015-0046-5. PMID: 26206151 Free PMC article.
fluorescence and solubility of 12 deletion mutants

Insertions and deletions in protein evolution and engineering.
Savino S, Desmet T, Franceus J. Biotechnol Adv. 2022 Jun 20;60:108010. doi: 10.1016/j.biotechadv.2022.108010. Online ahead of print.
Techniques of InDel construction

Minor points:
l. 169 Spelling: Escherichia coli
l. 169 suggested grammar: the region where the insertion is performed.
Figure 5. suggested grammar: ARG34 inserted in a beta sheet
Figure 5. indicate the color: Computational mutant (magenta) superimposed ...
l. 209 calculation of RDSM: The formula is not clear to me and reference 43 with potentially further information is not accessible (to me). Does the index vary from 1 to x or from 1 to LRC? What is w(x)? If x is constant for all i (i.e. w(LRC)) this term could be written outside the summation sign. Or does it vary with i - then why not write w(i)? What is WTi and Muti? The number of clusters with size i? This information should be made available. What happens, if WT and Mu do not have clusters of same size?

Author Response

Reviewer 3 Report (New Reviewer)

Amino acid insertions and deletions are often associated with diseases. However, despite numeric efforts to estimate the effects of non-frame-shift substitutions on protein structure or dynamics at a large scale, the study of frame shifted insertions or deletions has remained poorly explored.

In this manuscript, the authors developed a computational method that employed kinematic closure, structure relaxation, and rigidity analysis, to refine the inserted/deleted regions. The authors compared their method to mutated protein structures in PDB and showed that their method resulted in similar structures as experimentally resolved structures. The authors also proposed a rigidity score to estimate the stability of mutated proteins. The method seems to be powerful and applicable at a large scale. However, I have some concerns and comments before this manuscript can be accepted. 

1. The authors stated in the abstract that they can successfully model the effects of InDel mutations on protein structures. I would suggest the authors not to use such strong statement. Although the method can be potentially very useful, whether the method is actual successful is untested, since experimental and computational data for InDel mutations were limited.

2. The current organization of the manuscript could be improved. The authors may consider combining Figure 1 and 2, Figure 4 and 5, and Figure 6 and 7, respectively. 

Round 2

Reviewer 2 Report (Previous Reviewer 2)

I strongly support that section 3.8 was added. As far as this calculation is made comprehensible, the paper is acceptable. I have two requests in this respect:

It is not clear to me, how the fitness values were taken or calculated from Ref. 1 (Gonzales). I tried to calculate myself for Insertion 89,V and failed: I asume the numbers were calculated from the excel sheet "S1.Insertion ..." from the Supplement. Is that referring to the allele reported on line 1544 (Ambler Position 89)or line 1573 (Position 89)? Calculation of the fitness fi with formula (5) from the paper results in 2.7 and 2.6 resp., which corresponds to 6.5 and 6.1 ug/mL. How does this relate to the Fitness value of 0.257 in your table 2? This should be made clear in the Methods.

How did you select these 10 mutants out of the 4780 insertion mutants in sheet S1? Were they chosen randomly, if so in which way? The reader doesn't know. A different selection of 10 mutants might behave quite differently! It would definitely not be a proper procedure, if they were not chosen randomly.  

Minor points:
l. 9: insert "of": "the effects of InDels on protein fitness."

l. 82, 103 and other:  You use "modelling" 8 times and "modeling" 12 times. Be consistent.

l. 156: I would rather use "superimpose on" instead of "superimpose with".
  See also Fig. 3 (2x), l. 172

l. 170: "Escherichia coli" should be in italics.

l. 361: I would rather use "neither ... nor" instead of "not ... or" in the phrase "neither with respect to the RMSD nor (with respect) to the differences ...".

Author Response

This manuscript is a resubmission of an earlier submission. The following is a list of the peer review reports and author responses from that submission.

Round 1

Reviewer 1 Report

Amino acid insertions and deletions, or InDels, caused by non-frame shift genome variants, are commonly occurring in proteins. Jilani and coworkers here try to predict the structural impacts by these InDels. Specifically, in this study, the authors aim for understanding two things: 1, structural impact of the InDels occurring in secondary structure; 2) comparing the effects of InDels and substitutions.

Overall, this manuscript is very well-written. The major problem of this study is that there are not many PDB depositions with InDels on the protein (only 35), which limits the sample size of the investigation and jeopardizes the conclusion.

I only have a few minor edits.

Minor points

1.    Table 1.  Label the secondary structure, as either helix or strand, highlight some PDBs (bold fonts), which are specially discussed in the main text (for example, 2BBQ..).  

2.    Fgirue 4. Shows one example on the prediction when the InDels in the loop. The authors should also include one example when the InDels in secondary structure, since SS is the focus in this study.

3.      Page 11 line 264 - line 266. 2BBO (Cystic Fibrosis), 2NIP (Nitrogenase), and 1ANF (Maltodextrin binding protein), suggest the authors making a figure demonstrating these interesting results, since significant changes are observed.

4.      Section 3.5, Page 11, line 294-304. One of the focuses is the effects of the InDels in SS (secondary structure), the author should describe the results of finding on SS more thoroughly.

Author Response

Minor points

1. Table 1. Label the secondary structure, as either helix or strand, highlight some PDBs (bold fonts), which are specially discussed in the main text (for example, 2BBQ.).

Thank you so much for the suggestion. The proteins mentioned in the text have been highlighted in the table and secondary structure has been distinguished.

2. Figure 4. Shows one example on the prediction when the InDels in the loop. The authors should also include one example when the InDels in secondary structure, since SS is the focus in this study.

Great suggestion. The required example has been added as figure 5 in the manuscript. Thanks

3. Page 11 line 264 - line 266. 2BBO (Cystic Fibrosis), 2NIP (Nitrogenase), and 1ANF (Maltodextrin binding protein), suggest the authors making a figure demonstrating these interesting results, since significant changes are observed.

Another great suggestion. The results are interesting indeed, but we refrained from additional figures as it’ll be shifting the focus away from rigidity similarities to structural differences, which is core component of the paper. An additional example is added in figure 5 as kindly suggested.

4. Section 3.5, Page 11, line 294-304. One of the focuses is the effects of the InDels in SS (secondary structure), the author should describe the results of finding on SS more thoroughly.

This is a helpful suggestion. We did not see a big difference in secondary structures InDels versus loop InDels, but we only restricted ourselves to very short InDels. We will focus on it in future work. We added an explanation in the manuscript.

Reviewer 2 Report

The authors decribe a procedure to model the structural impact of InDels.

The main steps presented are modeling of InDel structures, rigidity analysis of the modeled structures and comparison of the rigidities.

For users it would be a great plus to apply the procedure fo her/his own projects. AlphaFold2 has had a big impact on modeling, due to its high quality. It was thus tempting to check if AlphaFold wouldn't suffice for the modeling step. With the few cases, modeling on a local installation of AlphaFold  gave models with global RMSDs (from SSM in coot) comparable to those given in Table 1. Thus, is there an advantage of the procedure applied in the paper or is the procedure just the result of pre-AlphaFold times.

The CGRAB server should be able to provide the rigidity analysis to any user.

The authors provide the proof of principle that a a computational model and its rigidity analysis can be used to estimate the impact of InDels. Is this result going beyond what one can test with existing webservices?

For sufficient impact of the paper, I would request either a more explicit (quantitative) answer to the question how bad or good the impact of an InDel on a protein is or an experimental test for the impact prediction. There should be enough examples of DeltaDeltaG values (reference protein vs. deletion mutant or reference protein vs. insertion mutant).

The procedure of rigidity analysis and it metrics are not common structural biology knowledge, yet. This part therefor needs more explanation. Part of this request examplifies in the l. 186 and l. 208 comments. On the other hand, the explanation of insertions and deletions I consider as textbook knowledge. This could be shortened.

Overall the structural impact of an InDel can be assessed, but for Biomolecules I expect more impact. The protein chemist wants to know, if an insertion mutant is worth to test in the lab, if it is expected to be more or less stable than the wild type. A more explicite answer in this respect will be worth publishing in Biomolecules.

Author Response

1. For users it would be a great plus to apply the procedure for her/his own projects. AlphaFold2 has had a big impact on modeling due to its high quality. It was thus tempting to check if AlphaFold wouldn't suffice for the modeling step. With the few cases, modeling on a local installation of AlphaFold gave models with global RMSDs (from SSM in coot) comparable to those given in Table 1. Thus, is there an advantage of the procedure applied int he paper or is the procedure just the result of pre-AlphaFold times.

Thank you for the great suggestion. We are considering to use AlphaFold information for the modeling part in future work. However, this requires careful consideration and testing, to see if it is also suitable for bigger Indels and not only local changes.

2. The CGRAB server should be able to provide the rigidity analysis to any user. The authors provide the proof of principle that a computational model and its rigidity analysis can be used to estimate the impact of InDels. Is this result going beyond what one can test with existing webservices?

We also model Indels, and not only provide rigidity analysis. It is the Indel modeling that proves to be the bigger challenge. The CGRAP server was developed by the authors of this paper for coarse-grained rigidity analysis, and it is part of the same research. However, the current version of CGRAP only performs the coarse grained analysis, and does not generate indel structures. Extension of CGRAP to generate indels and perform rigidity analysis on them is part of future work.

3. For sufficient impact of the paper, I would request either a more explicit (quantitative) answer to the question how bad or good the impact of an InDel on a protein is or an experimental test for the impact prediction. There should be enough examples of DeltaDeltaG values (reference protein vs. deletion mutant or reference protein vs. insertion mutant).The procedure of rigidity analysis and it metrics are not common structural biology knowledge, yet. This part therefore needs more explanation. Part of this request exemplifies in the l. 186 and l. 208comments. On the other hand, the explanation of insertions and deletions I consider as textbook knowledge. This could be shortened. Overall, the structural impact of an InDel can be assessed, but for Biomolecules I expect more impact. The protein chemist wants to know, if an insertion mutant is worth to test in the lab, if its expected to be more or less stable than the wild type. A more explicit answer in this respect will be worth publishing in Biomolecules.

Thank you for the suggestion. The metrics are indeed based on our previous work. We added a more extended explanation. 

Reviewer 3 Report

The authors study the effects of insertions and deletions on protein rigidity and compare effects of insertions and deletions to substitution mutations. The authors specifically focus on structures with indel mutations available in the PDB and compare the observed effects on rigidity between modelled mutants and the PDB structures of the mutants. The authors conclude that computationally generated indel mutations show similar properties to experimentally determined structures with these indels. I find that the manuscript doesn’t include any functional analysis and doesn’t make a good argument why rigidity as a metric would be of particular interest. I think it is imperative that the authors consider what effect these mutations might have on the functioning of these proteins. I also find the statistical basis for their claims lacking at present.

Major points:

1)    The authors focus on rigidity, but don’t explain why this would be significant for function. An indel might well have a larger effect on rigidity, but that doesn’t necessarily translate to any kind of biological effect, whereas specific mutations that conserve structural rigidity but change e.g. the electrostatic properties of binding sites might have a significant functional impact. 

2)    As the authors point out, many of these indels occur in distal loop regions, which again leads me to believe that they are mostly of marginal functional significance. 

3)    It is unclear to me what changes in rigidity (and their quantification as the RDSM score) tell me about the function of these biomolecules. Rigidity might be important for specific proteins, but the authors make no effort to illustrate how and why rigidity is important for the systems discussed in the manuscript. 

4)    No effort is made to obtain statistical confidence estimates for the rigidity calculations. Particularly for computationally generated models it should be easy to create a set of solutions and evaluate rigidity on all of them to determine the sensitivity of this calculation on minor structural variations. Hence, it is impossible to tell whether the changes displayed in figures 5,6 and 9 are statistically significant.

5)    The analysis of rigid clusters show in figures 7, 8 and 10 appear to be quite minor, also leading me to question whether any of these changes would have a significant functional effect

Minor points:

1)    The authors empathise the challenge of modelling indels. I don’t agree that this is such a unique challenge and homology modelling tools have addressed this problem satisfactorily for many years at this point.

Author Response

Major points:

1. The authors focus on rigidity, but don’t explain why this would be significant for function. An indel might well have a larger effect on rigidity, but that doesn’t necessarily translate to any kind of biological effect, whereas specific mutations that conserve structural rigidity but change e.g. the electrostatic properties of binding sites might have a significant functional impact.

Thank you for the comment. While we can measure the structural changes due to changes in rigidity, we cannot directly measure the impact on protein function. However, in previous work we were able to correlate the change in rigidity due to point substitutions to changes in delta-delta G. We expanded on it in the revised version.

2. As the authors point out, many of these indels occur in distal loop regions, which again leads me to believe that they are mostly of marginal functional significance.

Many Indels occur also in secondary structure elements, as is shown in our expanded dataset. It is indeed harder to obtain structural information for larger indels and/or indels in secondary structure

elements. We should add that Indels on loops may have significant functional effect, as loops are often involved in protein flexibility or binding. We clarified that in the revised manuscript.

3. It is unclear to me what changes in rigidity (and their quantification as the RDSM score) tell me about the function of these biomolecules. Rigidity might be important for specific proteins, but the authors make no effort to illustrate how and why rigidity is important for the systems discussed in the manuscript.

This is an important comment. As we explained above, we do not claim to directly measure the impact on protein function. We added a discussion about this in that section of the manuscript.

4. No effort is made to obtain statistical confidence estimates for the rigidity calculations. Particularly for computationally generated models it should be easy to create a set of solutions and evaluate rigidity on all of them to determine the sensitivity of this calculation on minor structural variations. Hence, it is impossible to tell whether the changes displayed in figures 5, 6 and 9 are statistically significant.

5. The analysis of rigid clusters shown in figures 7, 8 and 10 appear to be quite minor, also leading me to question whether any of these changes would have a significant functional effect.

Thanks for these important comments. Indeed it is true that we don’t explicitly perform statistical confidence calculations. However, the TLCCS metric (lines 204 – 215 in the original manuscript), which ranges from 0 to 1, provides a quantitative measure of the rigidity difference of the wildtype and mutant. In Table 1, TLCCS scores range from 0.3 to as high as 0.8. Thus, in some cases, the effects of the indels, as measured using the rigidity of the wildtype versus the rigidity of the mutant, are minor, but are significant in other cases.

Minor points:

1. The authors empathise the challenge of modelling indels. I don’t agree that this is such a unique challenge and homology modelling tools have addressed this problem satisfactorily for many years at this point.

See our response above. While it is possible to use homology modelling (which we will use in the future in the case of missing structures), our main goal was to analyze the effect of InDels on protein structure and test our ability to model the structure in a way that resembles the PDB structure.

Round 2

Reviewer 2 Report

I am still not recommending publication in Molecules.
As in the first report, the main point of critique leading to rejection is not the missing soundness of the manuscript. It is the missing expected impact. The modifications added by the authors do not principally change this. I still don't see, how the rigidity and its quatitative score RDSM relates to any measurable property, for instance temperature or denaturant stability.

The comments in Report 1 regarding the formulas (mainly l. 186 and l. 208 comments)might be irrelevant, but the authors have to reply to this point. Thus, my questions and requests pertain.

Reviewer 3 Report

I appreciate the revisions made to the manuscript text that qualifies the reliance on rigidity. 
However, I feel that the response to point 5 regarding statistical significance has not been addressed adequately. The authors have to show that their metrics are robust to minor variations in structure and that the observed differences hold up even if somewhat perturbed starting structures are employed. Such an analysis would allow the establishment of error estimates and rigorous significance testing, which in case it holds up, would strengthen the basis for the authors findings.